# Nano TiO_2_ and Molybdenum/Tungsten Iodide Octahedral Clusters: Synergism in UV/Visible-Light Driven Degradation of Organic Pollutants

**DOI:** 10.3390/nano12234282

**Published:** 2022-12-01

**Authors:** Margarita V. Marchuk, Igor P. Asanov, Maxim A. Panafidin, Yuri A. Vorotnikov, Michael A. Shestopalov

**Affiliations:** 1Nikolaev Institute of Inorganic Chemistry SB RAS, 3 Academician Lavrentiev Avenue, 630090 Novosibirsk, Russia; 2Boreskov Institute of Catalysis SB RAS, 5 Academician Lavrentiev Avenue, 630090 Novosibirsk, Russia

**Keywords:** TiO_2_, nanoparticles, octahedral iodide clusters, molybdenum, tungsten, rhodamine B, photodegradation, S-scheme heterojunction

## Abstract

Emissions of various organic pollutants in the environment becomes a more and more acute problem in the modern world as they can lead to an ecological disaster in foreseeable future. The current situation forces scientists to develop numerous methods for the treatment of polluted water. Among these methods, advanced photocatalytic oxidation is a promising approach for removing organic pollutants from wastewater. In this work, one of the most common photocatalysts—titanium dioxide—was obtained by direct aqueous hydrolysis of titanium (IV) isopropoxide and impregnated with aqueous solutions of octahedral cluster complexes [{M_6_I_8_}(DMSO)_6_](NO_3_)_4_ (M = Mo, W) to overcome visible light absorption issues and increase overall photocatalytic activity. XRPD analysis showed that the titania is formed as anatase-brookite mixed-phase nanoparticles and cluster impregnation does not affect the morphology of the particles. Complex deposition resulted in the expansion of the absorption up to ~500 nm and in the appearance of an additional cluster-related band gap value of 1.8 eV. Both types of materials showed high activity in the photocatalytic decomposition of RhB under UV- and sunlight irradiation with effective rate constants 4–5 times higher than those of pure TiO_2_. The stability of the catalysts is preserved for up to 5 cycles of photodegradation. Scavengers’ experiments revealed high impact of all of the active species in photocatalytic process indicating the formation of an S-scheme heterojunction photocatalyst.

## 1. Introduction

Currently, one of the most discussed problems in the world is the depletion of water resources and pollution of soils, water, and air [1,2]. These problems are associated with rapid industrialization and the growth of production capacities. The majority of plants consume enormous amounts of water and transform it into wastewater, which contains various inorganic and organic species. Approximately 80% of this wastewater is released into the environment without any prior treatment [3]. Among hazardous pollutants, organic dyes account for the largest share [4]. These substances are widely used in various industries, such as tanning, textile, paper, and printing industries, and can be considered one of the most harmful for the environment [5,6]. Besides high stability and such direct effects as carcinogenicity and mutagenicity [7], dyes are dangerous in long-term consequences due to high extinction. Higher absorption of colored water diminishes the amount of light in depth affecting the photosynthesis process and, henceforth, decreasing the overall amount of dissolved oxygen [8]. Modern methods for removal of such pollutants, such as coagulation, precipitation, and adsorption, have several disadvantages forcing researchers to search for alternative methods [9]. The major disadvantages of coagulation and precipitation are the generation of chemical sludge and the need for its subsequent treatment. In turn, the adsorption technique requires further regeneration of the adsorbent after utilization. In recent years, advanced oxidation processes for the photooxidation of organic compounds have attracted more and more attention as an alternative method for water treatment [10,11]. The photocatalysis process can be enabled by numerous compounds having semiconductor properties—organic [12,13,14] or inorganic [10,11]. Such a process has many advantages, such as complete mineralization of pollutants during the reaction, no need for regeneration after utilization, versatility, high efficiency even at low concentrations of pollutants, and compatibility with other methods [15]. For practical application, catalysts must meet a number of requirements: they should be chemically inert, stable, non-toxic, inexpensive, easy to manufacture, and have minimal human and environmental risks.

Titanium dioxide meets all the requirements and nowadays it is proven to be one of the most effective photocatalysts [16,17], taking into account numerous commercially available TiO_2_-based products [18]. Nevertheless, catalytic application of TiO_2_ under visible or sunlight is hindered due to its insufficient absorption in the visible spectrum (up to ~400 nm, E_g_ = 3.0–3.2 eV depending on the phase). This issue can be overcome by doping TiO_2_ with heteroatoms [19] or creating hybrid photocatalysts based on titanium dioxide and various organic or inorganic species, i.e., the creation of heterojunctions [20,21].

Octahedral cluster complexes of transition metals [{M_6_I_8_}L_6_]^n^, where M = Mo, W (Appendix A), are good candidates for combination with TiO_2_ to increase overall activity and expand absorption. Besides high chemical and photostability, high absorption, covering almost all visible range (up ~600 nm), and low optical band gap values (2–2.5 eV), clusters themselves possess photocatalytic activity. To date, photocatalytic dye degradation for the individual compound was shown only for the bromide cluster [{Mo_6_Br_8_}(N_3_)_6_]^2−^ [22]. Nevertheless, this complex also showed good activity being a component of a heterogenous catalyst based on gold nanoparticles and graphene oxide [23]. Also, it was shown, that various types of cluster-containing materials are efficient in the photocatalytic reduction of CO_2_ [24] or water splitting [25]. Concerning iodide molybdenum clusters, there is only one work, conducted in our group, on the synthesis of catalyst based on an almost inactive h-BN matrix [26]. In turn, tungsten iodide clusters are a relatively new class of compounds and, therefore, poorly studied from a practical point of view, i.e., there is no data on the photocatalytic activity of such complexes.

In this work, anatase-brookite mixed-phase nanosized particles were obtained via simple aqueous hydrolysis of titanium (IV) isopropoxide in a large excess of water. Water-soluble complexes [{M_6_I_8_}(DMSO)_6_](NO_3_)_4_ (M = Mo (1), W (2)) were utilized for the impregnation of titania powder. The dependency of composition, morphology, and efficiency of photocatalytic degradation of model dye Rhodamine B (RhB) under UV irradiation on the amount of cluster used for impregnation was studied. The most efficient catalysts were evaluated in cycling experiments and under direct sunlight irradiation. Experiments with scavengers for active species (h^+^, e^−^, OH^•^, and O_2_**^•^**^−^) were conducted to propose the mechanism of dye photodegradation.

## 2. Materials and Methods

### 2.1. Materials

Initial cluster complexes [{M_6_I_8_}(DMSO)_6_](NO_3_)_4_ (M = Mo or W) were synthesized according to the known procedure [27]. Other reagents and solvents were commercially available and were used without additional purification.

### 2.2. Instrumentation

Centrifugation was achieved by a Beckman Coulter Allegra X-30 centrifuge (Beckman Coulter, CA, USA) equipped with rotor F0630 (16,000 rpm, acceleration of 20,000 g). FTIR spectra were recorded as KBr pellets with a Bruker Vertex 80 spectrometer (Bruker Corporation, Billerica, MA, USA) from 400 to 4000 cm^−1^. The particle size and morphology were characterized by TEM with a Libra 120 microscope (Zeiss, Jena, Germany) at an acceleration voltage of 60 kV. Free image software “ImageJ” was used for particle size measuring. X-ray powder diffraction (XRPD) patterns were recorded on a Shimadzu XRD 7000S diffractometer (Shimadzu Corporation, Kyoto, Japan) (Cu Kα radiation, graphite monochromator, and silicon plate as an external standard). The size of the crystallites was calculated from XRPD data using the Scherrer equation:(1)$$D=\frac{K\times \lambda }{\beta \times \cos\theta }$$
where D is crystallites size in nm, K is Scherrer constant and equal to 0.9, λ is the wavelength of the X-ray source in nm (Cu Kα = 0.15406 nm), β is full width at half maximum of the peak in radians, Θ is peak position in radians.

Metal content was determined using inductively coupled plasma atomic emission spectroscopy (ICP-AES) on a ThermoScientific spectrometer iCAP-6500. The relative error is about 5–10%. Diffuse reflectance spectra were recorded using a UV-Vis-NIR 3101 PC spectrophotometer (Shimadzu Corporation, Kyoto, Japan). The emission spectra (λ_ex_ = 320 nm) were recorded for powdered materials placed between two non-fluorescent glass plates with Agilent Cary Eclipse Fluorescence Spectrophotometer.

X-ray photoelectron spectroscopy (XPS) and valence band XPS (VB XPS) was performed on X-ray photoelectron spectrometer SPECS (SPECS Surface Nano Analysis GmbH, Berlin, Germany) using nonmonochromatic Al Kα irradiation (hν = 1486.6 eV). A thin layer of the powdered sample was applied onto conductive double-sided copper tape (3M^™^, Electron Microscopy Sciences, Hatfield, PA, USA). The binding energy scale was preliminarily calibrated by the position of the photoelectron lines of the core levels of gold (Au4f_7/2_, 84.0 eV), silver (Ag3d_5/2_, 368.3 eV), and copper (Cu2p_3/2_, 932.7 eV). To calibrate the recorded spectra, the Ti2p_3/2_ line (BE = 458.8 eV) from the TiO_2_ matrix was used as an internal standard.

An analysis of the porous structure was performed by a nitrogen adsorption technique using Quantochrome’s Autosorb iQ at 77K. Initially, the materials were activated in a dynamic vacuum at 200 °C for 2 h. N_2_ adsorption-desorption isotherms were measured within the range of relative pressures of 10^−6^ to 0.995. The specific surface area was calculated from the data obtained on the basis of the conventional BET and DFT models. Pore size distributions were calculated using the DFT method.

UV-light irradiation in photodegradation experiments was performed with a Hamamatsu Photonics light-emitting-diode (LED) head unit L11921-400 (wavelength 365 ± 5 nm, ~13 mW cm^−2^) used with a LED controller C11924-211 (Hamamatsu Photonics, Hamamatsu City, Japan). Absorption spectra were recorded on an Agilent Cary 60 spectrophotometer (Agilent, Santa Clara, CL, USA).

### 2.3. Synthesis of Pure TiO_2_

The titanium dioxide was obtained by the hydrolysis of titanium (IV) isopropoxide (Ti(^i^OPr)_4_) [28,29,30]. The titanium (IV) isopropoxide was added dropwise to the hot distilled water (80 °C) at vigorous stirring in a 1:50 volume ratio. The resulting suspension was immediately sonicated for 5 min and then kept stirred at 80 °C for 4 h. Resulting white powder was washed 3 times with water and 2 times with acetone. The pure TiO_2_ was dried in air at room temperature. Yield: ~100%.

### 2.4. Synthesis of n^x^@TiO_2_ (n = 1 (Mo) or 2 (W))

Cluster-containing materials n^x^@TiO_2_ (where *n* = 1 (Mo) or 2 (W); x is the mass of the complex in gram per 1 g of TiO_2_ and equal to 0.1, 0.5, 1, and 1.5) were obtained by impregnation of TiO_2_ with the solution of cluster complex [{M_6_I_8_}(DMSO)_6_](NO_3_)_4_ (M = Mo or W) in water. In a typical experiment, the required amount of the corresponding cluster was dissolved in 100 mL of water and 1 g of pure TiO_2_ was added to the solution at constant vigorous stirring. The resulting suspensions were stirred for 20 h. Light yellow powders of n^x^@TiO_2_ were washed several times with water until the washings became colorless and then 3 times with acetone. Materials were dried in air at room temperature.

### 2.5. Photocatalytic Experiment

Photocatalytic activity of n^x^@TiO_2_ materials was investigated in the photodegradation of Rhodamine B (RhB) under UV irradiation (λ = 365 ± 5 nm, ~13 mW cm^−2^). In a typical photocatalytic test, 20 mg of n^x^@TiO_2_ and 60 mL of water were mixed in a quartz reactor and sonicated for 5 min. Then 20 mL of RhB solution (C = 10 mg L^−1^) was added to the reactor and the resulting mixture was stirred for 2 h in the dark to reach adsorption-desorption equilibrium. The overall volume is 80 mL, RhB concentration is 2.5 mg L^−1^, n^x^@TiO_2_ concentration is 0.25 g L^−1^. After that, the reaction mixture was irradiated with UV light for 45–120 min depending on the photocatalytic activity of the sample. During the irradiation, several aliquots of solutions (8 mL) were collected and centrifuged. UV-vis spectra of the isolated solutions were recorded in order to estimate the concentration of RhB. The decrease in the RhB concentration was tracked by its characteristic absorbance at 554 nm. The reactions rate constants (k_eff_) were determined as pseudo-first order kinetics by linear approximation of ln(C/C_0_) vs. t plot where C is the concentration of RhB at corresponding t, C_0_ is the initial concentration of RhB, and t is the time when aliquots of solutions were taken. The effect of catalyst concentration was studied using 10 or 30 mg of n^0.1^@TiO_2_ (n^0.1^@TiO_2_ concentrations are 0.125 or 0.375 g L^−1^).

### 2.6. Cyclic Experiments

Cyclic experiments were performed in a similar manner as a typical photocatalytic test. After each run of irradiation (60 min) the aliquot (8 mL) was taken, centrifuged, and the UV-vis spectrum of the isolated solution was recorded. The remaining precipitate was placed back into the reaction mixture and the initial concentration of RhB (2.5 mg L^−1^) and overall volume (80 mL) were adjusted by the addition of an RhB solution.

### 2.7. Photodegradation of RhB under Solar Light

To study photodegradation under sun irradiation, a typical photocatalytic experiment was performed on a clear day in the middle of august (sunlight power = ~30–35 mW cm^−2^). The reaction mixture in the quartz reactor without stirring was exposed to solar irradiation for 90–120 min depending on the photocatalytic activity of the sample. The air temperature was ~25 °C.

### 2.8. Scavengers

Photocatalytic activity of n^0.1^@TiO_2_ was assessed using the same procedure as earlier but in presence of scavengers for active species: Na_2_C_2_O_4_ (h^+^, C = 10 mM), AgNO_3_ (e^−^, C = 10 mM), i-PrOH (OH^•^, C = 10 mM). For the determination of O_2_**^•^**^−^ impact the reaction mixture was preliminarily deaerated by bubbling with Ar gas for 20 min. Relative activity (RA) was calculated according to the following equation:(2)$$\mathrm{RA}=\frac{k_{\mathrm{eff}}\left(\mathrm{scav}\right)}{k_{\mathrm{eff}}\left(\mathrm{NS}\right)}\times 100\%,$$
where k_eff_(scav) is an effective rate constant in the presence of a certain scavenger and k_eff_(NS) is an effective rate constant in a scavenger free experiment.

## 3. Results and Discussion

### 3.1. Synthesis of Pure TiO_2_ and n^x^@TiO_2_ (n = 1 (Mo) or 2 (W))

Titanium dioxide was obtained by a simple hydrolysis of titanium (IV) isopropoxide in a neutral aqueous medium, i.e., by addition of Ti(^i^OPr)_4_ into great excess of water (volume ratio is 1:50). For additional homogenization the resulting dispersion was sonicated immediately after the addition of titanium precursor. To transform partially amorphous TiO_2_ into a crystalline phase the dispersion was heated at 80 °C for 4 h. According to XRPD analysis (Figure 1G), the resulting TiO_2_ powder is predominantly anatase phase with the admixture of brookite. The broadening of the peaks in the diffractogram indicates the formation of small particles. Using Scherrer Equation (1) we calculated the average size of TiO_2_ particles, which is equal to 4.3 ± 1.0 nm. High dilution of the reaction mixture and low temperature probably results in the formation of extremely small crystalline mixed-phase particles.

Cluster-containing materials n^x^@TiO_2_ (n = 1 (Mo) or 2 (W); x is the mass of the complex in gram per 1 g of TiO_2_ and equal to 0.1, 0.5, 1, and 1.5) were obtained by impregnation of TiO_2_ dispersion with a certain amount of cluster complexes [{Mo_6_I_8_}(DMSO)_6_](NO_3_)_4_ (1) or [{W_6_I_8_}(DMSO)_6_](NO_3_)_4_ (2) in water. XRPD analysis of the resulting yellowish powders demonstrates an absence of cluster-related peaks, indicating deposition of complex in amorphous state regardless of composition of cluster core (Figure 1G and Appendix A). Based on our experience and reaction conditions we believe that cluster is deposited on the TiO_2_ surface in the form of fully hydrolyzed neutral aqua-hydroxo complexes [{M_6_I_8_}(H_2_O)_2_(OH)_4_]·nH_2_O. The absence of any ligand-related peaks in FTIR spectra also confirms the full hydrolysis of the clusters (Appendix A). The particle size of n^x^@TiO_2_ calculated from diffractogram is close and almost independent of the amount of complex nor the type of cluster core, but slightly higher, than that of native TiO_2_, 4.3 ± 1.0 nm (TiO_2_) vs. 4.8 ± 0.7 (1^0.1^@TiO_2_) and 4.7 ± 0.8 (2^0.1^@TiO_2_) nm.

### 3.2. Morphology and Composition

The morphology of the initial TiO_2_ and n^x^@TiO_2_ was studied using TEM (Figure 1A–C). One can see that all of the samples consist of small, uniformly distributed particles forming agglomerates of ~100 nm or higher in diameter, indicating no influence of cluster impregnation on the morphology of the TiO_2_. Statistical analysis of the particle size (Figure 1D–F) results in normal distribution curves with maxima of 4.8 ± 0.9 nm for TiO_2_, 5.0 ± 0.8 nm for 1^0.1^@TiO_2_, and 7.7 ± 1.3 nm for 2^0.1^@TiO_2_, which is close to the data from XRPD. The content of the molybdenum or tungsten in n^x^@TiO_2_ was determined using ICP-AES and then converted to the content of {M_6_I_8_} per 1 g of TiO_2_ (Figure 1H, Appendix A). According to the data obtained, the molybdenum cluster impregnates TiO_2_ far less than the tungsten one. Even at x = 0.1, the content of {W_6_I_8_} is in order of magnitude higher than the content of {Mo_6_I_8_}, which is in agreement with the larger particle size of 2^0.1^@TiO_2_, determined from TEM. The subsequent increase in molybdenum cluster complex loading (x) does not lead to the increase in the {Mo_6_I_8_} content in 1^x^@TiO_2_. On the contrary, the increase in the loading of the tungsten complex results in a linear increase in the {W_6_I_8_} content in the 2^x^@TiO_2_ materials. Such difference in the interaction of the clusters with TiO_2_ is most likely due to the different behavior of the clusters during hydrolysis i.e., related to the form of the cluster in the solution [27,31]. [{Mo_6_I_8_}(DMSO)_6_](NO_3_)_4_, being dissolved in water, quickly hydrolyzes with the formation of water-soluble cationic complexes [{Mo_6_I_8_}(H_2_O)_6−m_(OH)_m_]^(4−m)+^ (m = 0–3). In turn, tungsten complex hydrolyzes much slower than molybdenum one and exists in solution predominately in non- or partially hydrolyzed forms. We believe that cationic complexes [{Mo_6_I_8_}(H_2_O)_6−m_(OH)_m_]^(4−m)+^ (m = 0–3) interact well with surface OH- and H_2_O groups of TiO_2_, forming direct covalent bonds Ti-O-Mo or hydrogen bonds Ti-O(H)···(H)O-Mo, or both, and do not interact with already deposited cluster units, thus forming a monolayer on the particle surface. Thus, almost complete saturation of the TiO_2_ surface was achieved even at the lowest concentration of molybdenum cluster in solution (x = 0.1) and the following increase in the cluster concentration does not result in the increase in impregnation degree. On the contrary, partially hydrolyzed forms of tungsten cluster apparently interact with other cluster units via covalent or hydrogen bonds more effectively resulting in the growth of a “cluster shell” on the surface of the TiO_2_ particles.

### 3.3. X-ray Photoelectron Spectroscopy (XPS)

X-ray photoelectron spectroscopy (XPS) is one of the best methods for studying the composition and the valence states of the elements in bulk materials [26,32,33,34,35,36], and therefore it was used for investigation of n^0.1^@TiO_2_ (n = 1 or 2) (Figure 2 and Appendix A). A high noise level was observed for all cluster-related peaks, which is due to the low content of the complexes (Figure 2A,B,D,E). Nevertheless, metals peak positions, i.e., Mo3d_5/2_ (228.5 eV), Mo3d_3/2_ (231.7 eV), and W4d_5/2_ (243.9 eV), W4d_3/2_ (256.7 eV), correspond to M^2+^ state, which is characteristic for cluster complexes. Iodine signals, besides main peaks at I3d_5/2_ (620.1 eV) and I3d_3/2_ (631.6 eV), contain shoulders at I3d_5/2_ (618.3 eV) and I3d_3/2_ (629.7 eV), which can be attributed to apical I^–^ ligands unsubstituted during the synthesis of initial clusters (~1 apical I^–^ per {M_6_I_8_}) [26]. Importantly, the type of the metal in the cluster core does not affect the position of iodine ligands peaks. Spectra of titanium and oxygen are identical for both types of materials (Figure 2C,F and Appendix A). Ti2p_3/2_ and Ti2p_1/2_ peaks are located at ∼458.8 and 464.5 eV, which is in agreement with the literature data for Ti^4+^ in TiO_2_ (Figure 2C and Appendix A) [32,33]. In the O1s spectrum, one can see three components located at 530.1, 531.5, and 532.7 eV (Figure 2F and Appendix A). The most intensive peak (530.1 eV) can be attributed to lattice oxygen atoms in the TiO_2_ [34,35]. The other two components (531.5 and 532.7 eV) can be attributed to surface groups—bridging oxygen and -OH groups, respectively [34,35]. Thus, no specific signals related to the binding of cluster complexes with surface groups of TiO_2_ were observed. Nevertheless, the data obtained do not refute the formation of covalent M-O-Ti and/or hydrogen bonds M-OH/H_2_O···HO-Ti, especially taking into account the low content of the clusters in the material.

### 3.4. Surface Area and Porous Structure

Measured isotherms of nitrogen adsorption at 77 K are represented in Figure 3. All of the materials studied are possessing formally type II isotherm according to the official IUPAC classification [37], which is typical for non-porous or macroporous compounds, for which unrestricted monolayer-multilayer adsorption can occur. The inflection point (point B) lies at quite low relative pressures *P*/*P*_0_ (less than 2·10^−3^) indicating the presence of micropores, though their volume is not high due to low volume adsorbed at this pressure (less than 40 mL g^−1^). At high relative pressures, hysteresis loops of H3 type are observed for all studied materials. Such a type of hysteresis is typically associated with an irregular pore network consisting of meso- and macropores that are not completely filled with condensate.

The specific surface area was calculated by the conventional BET method and DFT approach. Pore volumes were calculated using the DFT method. All calculated parameters of the porous structure are given in Table 1.

The pore size distribution plot (inset in Figure 3) shows the presence of irregular mesopores with a wide maximum with a diameter of 6–8 nm. One (TiO_2_) or two (n^0.1^@TiO_2_) peaks in the microporous range in distribution graphs are attributed to narrow pores but their contribution to overall porosity is negligible (less than 0.02 mL g^−1^ from about 0.4 mL g^−1^). One can note that pore volume is diminishing after impregnation with clusters, and V_pore_ value of 2^0.1^@TiO_2_ is lower than that of 1^0.1^@TiO_2_, which correlates well with ICP-AES data, i.e., the higher the content of the cluster, the lower the volume of pores.

### 3.5. Absorption and Emission

Absorption of pure TiO_2_ and n^0.1^@TiO_2_ was studied by diffuse-reflectance spectroscopy. The spectra obtained were transformed using the Kubelka-Munk function (Figure 4A). According to the data, impregnation with cluster does not result in a dramatic increase in absorption in the visible range. However, a small shoulder up to ~500 nm, which forms and intensity is independent of the type of the cluster, can be observed in the enlarged part of Figure 4A. Despite the different content of the complexes, the overall impact of cluster impregnation on the absorption of the materials is the same. This is due to the difference in absorption of the clusters themselves–molybdenum clusters have wider absorption than tungsten ones, but a smaller degree of {Mo_6_I_8_} impregnation equalizes the overall effect. In turn, the effect of cluster content is appearing in the UV region, since 2^0.1^@TiO_2_ has higher than 1^0.1^@TiO_2_ absorption in 250–350 nm range.

To measure optical band gap (E_g_), the Tauc plot was used. The dependence of (αhν)12 versus hν was plotted and interception of straight lines with y = 0 gives E_g_ values (Figure 4B). Despite the admixture of brookite phase, pure TiO_2_ and both cluster-containing materials demonstrates typical for anatase band gap value of 3.2 eV [17]. Interestingly, but on the spectra of both n^0.1^@TiO_2_ one can see another straight line with E_g_ of 1.8 eV, which can be attributed to the shoulder in the absorption spectra (Figure 4A). This value can be referred to hydrolyzed cluster complex on the surface of the particles (e.g., E_g_ of [{Mo_6_I_8_}(H_2_O)_2_(OH)_4_]·2H_2_O is 1.7 eV [38]), thus confirming our earlier suggestion.

Emission spectroscopy is useful tool to estimate the separation efficiency and transfer ability of photogenerated charge carriers since electron-hole pairs recombination results in emission. Emission spectra of pure TiO_2_ and n^0.1^@TiO_2_ were recorded for the samples in solid state (Figure 4C). Both n^0.1^@TiO_2_ demonstrates similar intensity of the emission which is lower than the intensity of pure TiO_2_ for ~25%. Lowering of the emission intensity indicates higher lifetime of electron-hole pairs. Note, that there is no cluster originated emission on the spectra indicating effective charge carriers transfer between complex and matrix.

### 3.6. Photocatalytic Degradation of RhB

*UV-light.* The photocatalytic activity of the samples was evaluated under UV-light irradiation (λ = 365 nm, 13 mW cm^−2^). Rhodamine B (RhB) was chosen as the model organic pollutant (Appendix A). Before carrying out photocatalytic experiments, the reaction mixture was kept stirred in dark for 2 h to achieve sorption-desorption equilibrium. After that, the mixture was irradiated with UV light under constant stirring, and aliquots were taken in different time intervals. The dye degradation rate was monitored on UV-vis spectra by reduction in characteristic RhB band at 554 nm (Figure 5). Effective rate constants were calculated from linear approximation of ln(C/C_0_) vs. t plots (where C is concentration of RhB at corresponding t, C_0_ is initial concentration of RhB, t is time when aliquots were taken) (inserts in Figure 5) and presented in Table 2. According to the data obtained, pure TiO_2_ does not degrade dye completely even after 90 min of irradiation and its k_eff_ is 0.02 min^−1^. In turn, both n^0.1^@TiO_2_ demonstrate close activity completely decomposing RhB after 45 min with k_eff_ of ~0.099–0.11 min^−1^, which is ~5 times higher than that of titania nanoparticles. Taking into account experimental conditions, rate constant values of n^0.1^@TiO_2_ are comparable to the most efficient photocatalysts known in literature [39,40]. Concerning the dependency of n^x^@TiO_2_ efficiency on cluster content, one can see, that k_eff_ behaves differently for materials containing molybdenum or tungsten clusters (Table 2). All of the 1^x^@TiO_2_ demonstrate similar activity with k_eff_ of ~0.1 min^−1^, while for 2^x^@TiO_2_ k_eff_ gradually decreases from 0.11 (for x = 0.1) to 0.025 (for x = 1.5) min^−1^. This tendency correlates well with the real cluster content determined using ICP-AES (Figure 1H), i.e., in 1^x^@TiO_2_ the content of {Mo_6_I_8_} is independent on x, thus here we have similar activity of all of the samples. In turn, the content of {W_6_I_8_} in 2^x^@TiO_2_ increases with the increase in the amount of complex in reaction mixture, and k_eff_ depends linearly on cluster core content (Appendix A). This behavior is probably associated with shielding of the titanium dioxide surface when a large amount of the complex is deposited, due to which TiO_2_ does not participate in the photocatalytic reaction, thus the material exhibits mostly the activity of cluster. This phenomenon demonstrates well the synergistic effect in the titled system. The effect of catalyst concentration was studied using 10 or 30 mg of n^0.1^@TiO_2_ per 80 mL of the reaction mixture (C_cat_ are 0.125 or 0.375 g L^−1^). According to the data obtained, the changes in catalyst concentrations resulted in a decrease in efficiency (Table 2), which makes a concentration of 0.25 g L^−1^ the most optimal. Since the highest catalytic efficiency under UV light along with the lowest cluster content was shown for n^0.1^@TiO_2_, these samples were chosen for further studies.

*Cyclic experiments.* To study the stability of the materials in photocatalytic reactions cyclic experiments were conducted. n^0.1^@TiO_2_ samples were chosen due to highest activity at lowest cluster content. In general, 5 runs of irradiation were performed. To avoid the accumulation of oxidized byproducts, which can decrease the rate of the reaction, irradiation time was increased to 60 min. After each run, the aliquot was taken and the concentration of RhB and n^0.1^@TiO_2_ was adjusted to the initial state. The results obtained are presented in Figure 6, and one can see, that both of the samples do not loss their activity up to 5 cycles.

*Solar light.* In addition, photocatalytic activity of pure TiO_2_ and n^0.1^@TiO_2_ was evaluated under direct sunlight irradiation. The experiments were performed as earlier, but without stirring in the middle of august under clear sky (T_air_ ~25 °C). The complete decomposition of RhB in the presence of pure titania was achieved in 90 min with k_eff_ of 0.036 min^−1^ (Figure 7). Similar to experiments under UV-light, both n^0.1^@TiO_2_ demonstrated close activity decomposing RhB after ~45–50 min with k_eff_ of 0.12 and 0.14 min^−1^ for n = 1 and 2 correspondingly. One can see, that all of the samples show some increase in the activity, which is due to the continuous spectrum of solar light providing more photons with a suitable wavelength.

### 3.7. Scavengers

Scavengers are compounds that can selectively react with certain active species, e.g., OH^•^, O_2_**^•^**^−^, e^−^, or h^+^. Decrease in the rate of the photocatalytic reaction in the presence of such compounds indicates the impact of specific active species, which is helpful for revealing mechanism of photocatalytic activity. Here we used ^i^PrOH, AgNO_3_, and Na_2_C_2_O_4_ as specific scavengers for OH^•^, e^−^, and h^+^, respectively. To study the effect of O_2_**^•^**^−^ the reaction mixture was deaerated by the bubbling of Ar gas. The photocatalytic experiments were conducted as earlier and k_eff_ in the presence of scavengers were calculated (Figure 8A,B, Appendix A). Using obtained k_eff_ values, corresponding relative activities for each type of the scavenger were calculated according to Equation (2) (Figure 8C).

One can see that all of the scavengers used in certain degree reduce the activity of the catalysts, indicating impact of all active species on photocatalytic process. The lowest effect was achieved in the presence of ^i^PrOH (decrease in activity for 11 and 13% for 1^0.1^@TiO_2_ and 2^0.1^@TiO_2_ correspondingly), therefore OH^•^ is the least active particle for both type of catalysts. Scavenging of e^−^ and h^+^ resulted in greater activity lowering—for 81 and 72% for 1^0.1^@TiO_2_ and for 67 and 57% for 2^0.1^@TiO_2_. The highest inhibition was achieved in deaerated atmosphere (decrease in activity for 85 and 91% for 1^0.1^@TiO_2_ and 2^0.1^@TiO_2_ correspondingly) proving that⋅O_2_**^•^**^−^ are the dominant active species in photodegradation of RhB.

### 3.8. Photocatalytic Mechanism

Thus, participation of all active species—h^+^, e^−^, OH^•^ and O_2_**^•^**^−^, in photocatalytic process along with significantly higher activity of the material compared with pure TiO_2_ allow us to suggest the formation of S-scheme (also known as direct Z-scheme) heterojunction in n^0.1^@TiO_2_ [41]. In such systems, due to the formation of heterojunction, resulting in band curving, and activity of both components under irradiation, less active electrons and holes can recombine thereby preserving electrons and holes having higher energy. In order to illustrate band positions in the materials, E_VB_-E_F_ (E_VB_ is the edge positions of valence bands, E_F_ is Fermi level potential) distances in n^0.1^@TiO_2_ were determined using valence band XPS (VB-XPS) (Figure 9). One can see on the Figure 9A, that intercept of straight lines gives us value of 2.86 eV, which is in good agreement with literature data for TiO_2_ [42]. Nevertheless, similar to absorption graphs, there are additional shoulders, which gives E_VB_-E_F_ distances of 0.25 eV independently to the type of the metal in cluster core (Figure 9B). These values can be attributed to the presence of clusters in the materials.

According to literature data, the valence band potential (E_VB_) of TiO_2_ is −7.55 eV vs. E_VAC_ (3.05 eV vs. NHE) [43,44,45]. Thus, the conduction band potential (E_CB_) of titania was calculated using equation $E_{g}=E_{\mathrm{VB}}-E_{\mathrm{CB}}$ and was found to be −4.35 eV vs. E_VAC_ (0.05 eV vs. NHE). Fermi level position (E_F_) was calculated using E_VB_-E_F_ distance (2.86 eV) and was found to be −4.69 eV vs. E_VAC_ (0.19 eV vs. NHE). Assuming that when the semiconductors are in contact, their Fermi levels are aligned, while the positions of the valence and conduction bands (VB and CB) remain unchanged, we calculated E_VB_ of clusters, which are equal to −4.94 eV vs. E_VAC_ (0.54 eV vs. NHE), and then the position of the conduction band—−3.14 eV vs. E_VAC_ (−1.36 eV vs. NHE). Therefore, using the data obtained we can propose the band gap structure and mechanism of the RhB photodegradation (Figure 10). Thus, according to calculations, the potentials obtained are sufficient for the generation of all radicals, which is in agreement with scavenger experiments and overall confirms the formation of S-scheme heterojunctions.

The mechanism of dye photodegradation can be depicted according to Equations (3)–(7). First, both components absorb photons producing electron-hole pairs (Equation (3)). Next, the electron on TiO_2_ and the hole on the cluster, having low potentials, recombine (Equation (4)). In turn, the electron on the cluster and the hole on TiO_2_, preserved due to recombination of other pair, have sufficient potential to produce O_2_^•−^ or OH^•^ (Equations (5) and (6)). Then all active species formed mineralize RhB molecule (Equation (7)).
cluster@TiO_2_ + hν → cluster(h^+^ + e^−^)@TiO_2_(h^+^ + e^−^)(3)
cluster(h^+^) + TiO_2_(e^−^) → cluster@TiO_2_ (recombination)(4)
cluster(e^−^) + O_2_ → O_2_^•−^(5)
TiO_2_(h^+^) + H_2_O/OH^−^ → OH^•^(6)
cluster(e^−^)/TiO_2_(h^+^)/O_2_^•^/OH^•^ + RhB → … → CO_2_ + H_2_O + etc(7)

## 4. Conclusions

To conclude, anatase-brookite mixed-phase TiO_2_ nanoparticles were obtained by an aqueous hydrolysis of titanium (IV) isopropoxide in a large excess of water. Impregnation of the titania with a varied amount of [{M_6_I_8_}(DMSO)_6_](NO_3_)_4_ (M = Mo (1), W (2)) resulted in the formation of hybrid cluster-containing photocatalysts n^x^@TiO_2_ (n is Mo or W cluster, x is the amount of cluster used in impregnation reaction). Analysis of the composition revealed a strong effect of the type of the cluster on impregnation rate: more hydrolytically unstable complex 1 impregnates TiO_2_ surface in equal amount regardless of x, while in the case of more stable complex 2 the content of {W_6_I_8_} grows linearly with x. Cluster deposition results in moderate increase in the absorption in visible range up to ~500 nm and appearance of additional cluster-related band gap with value of 1.8 eV. Photocatalytic activity of the samples in RhB degradation is directly affected by the amount of the cluster deposited on the particle—all 1^x^@TiO_2_ demonstrates similar activity, and for 2^x^@TiO_2_ decrease in the efficiency along with x was observed. Thus, highest efficiency was achieved for n^0.1^@TiO_2_ with rate constants of ~0.1–0.14 min^−1^ under both UV- and sunlight irradiation, which is 4–5 times higher than those of pure TiO_2_. Both the materials are stable for at least for 5 cycles of photocatalytic degradation of the dye without any loss in the efficiency. High activity of the materials comparing to pure TiO_2_ along with impact of all of the active species, revealed in scavengers’ experiments, indicates formation of S-scheme heterojunction catalysts. Thus, the materials obtained here demonstrate high efficiency both under UV and sunlight irradiation and, therefore, they can be used for efficient wastewater treatment from organic pollutants.

## Figures and Tables

**Figure 1 nanomaterials-12-04282-f001:**
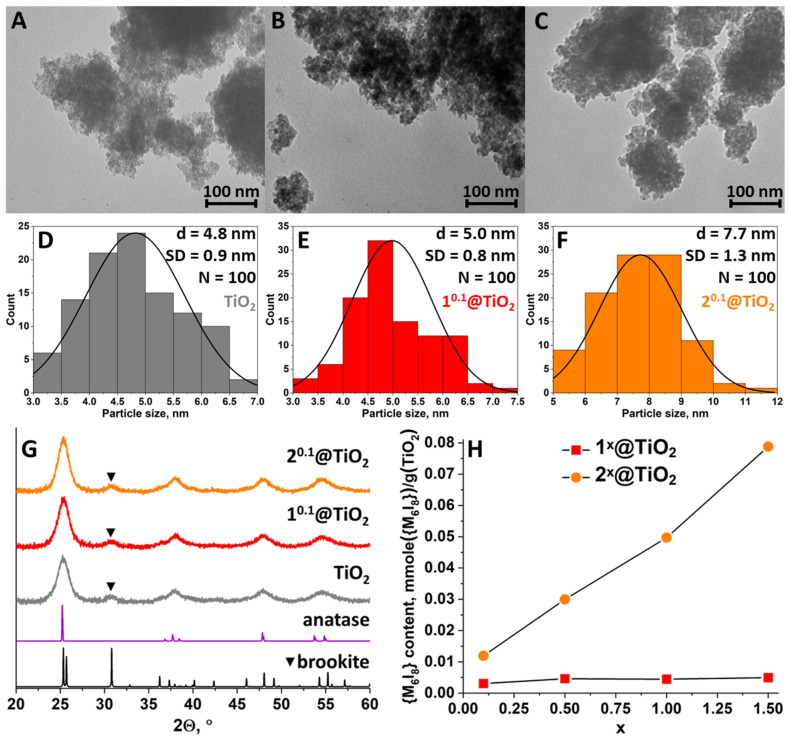
(**A**–**C**) TEM images of TiO_2_ and n^0.1^@TiO_2_ (n = 1 (Mo) and 2 (W)); (**D**–**F**) Particle size analysis using TEM images; (**G**) XRPD patterns of TiO_2_ and n^0.1^@TiO_2_ in comparison with calculated diffractogram of anatase and brookite (triangles refer to brookite phase reflection); (**H**) Content of {M_6_I_8_} cluster core per 1 g of TiO_2_ in n^x^@TiO_2_ determined using ICP-AES.

**Figure 2 nanomaterials-12-04282-f002:**
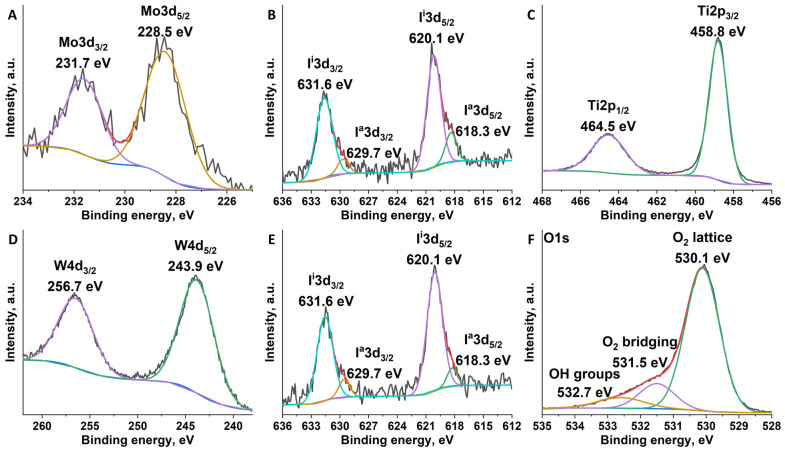
High-resolution XPS spectra of Mo3d (**A**), I3d (in 1^0.1^@TiO_2_) (**B**), Ti2p (in 1^0.1^@TiO_2_) (**C**), W4d (**D**), I3d (in 2^0.1^@TiO_2_) (**E**), and O1s (in 1^0.1^@TiO_2_) (**F**) core levels in n^0.1^@TiO_2_.

**Figure 3 nanomaterials-12-04282-f003:**
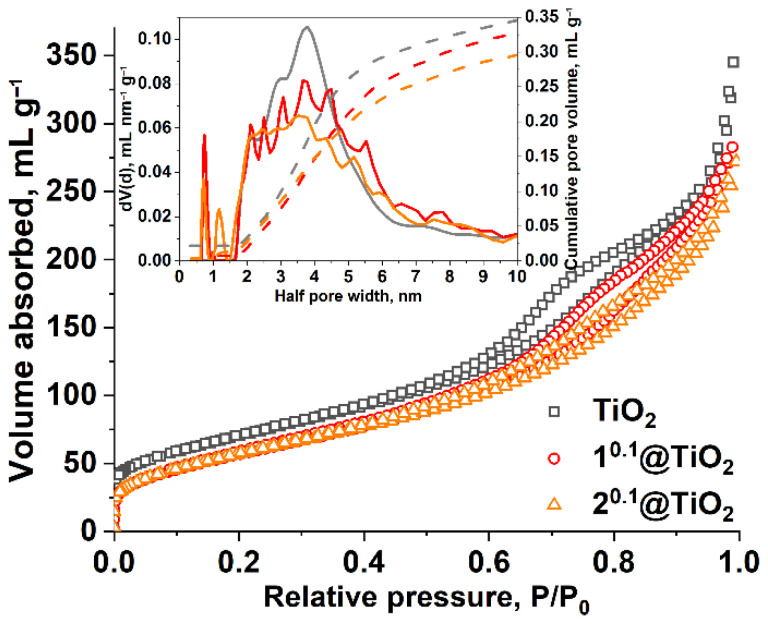
Nitrogen adsorption-desorption isotherms at 77 K. Inset shows pore sizes distribution according to DFT calculations. Color code: gray–TiO_2_, red–1^0.1^@TiO_2_, orange–2^0.1^@TiO_2_.

**Figure 4 nanomaterials-12-04282-f004:**
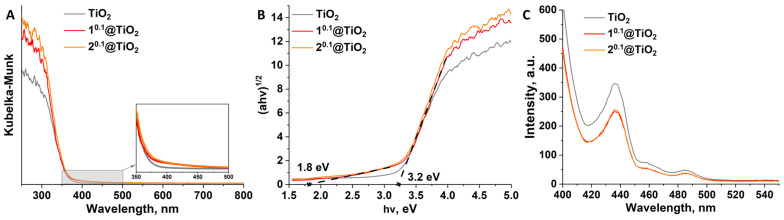
(**A**) Diffuse reflectance spectra of pure TiO_2_ and n^0.1^@TiO_2_ converted to absorption spectra using the Kubelka-Munk function; (**B**) Tauc plots used for determination of E_g_. Dashed lines are approximations of linear parts of the spectra. X marks interception with x axis and corresponds to E_g_ values; (**C**) Emission spectra of powdered TiO_2_ and n^0.1^@TiO_2_ (λ_ex_ = 320 nm).

**Figure 5 nanomaterials-12-04282-f005:**
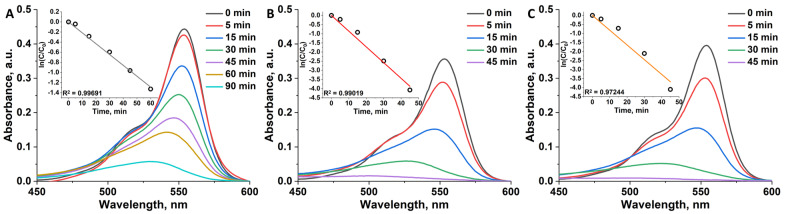
Absorption spectra of RhB solution before and after irradiation with UV light (λ = 365 nm, 13 mW cm^−2^) in the presence of TiO_2_ (**A**), 1^0.1^@TiO_2_ (**B**), and 2^0.1^@TiO_2_ (**C**) during different time intervals. Inserts are linear approximation of ln(C/C_0_) vs. time plots used for determination of k_eff_.

**Figure 6 nanomaterials-12-04282-f006:**
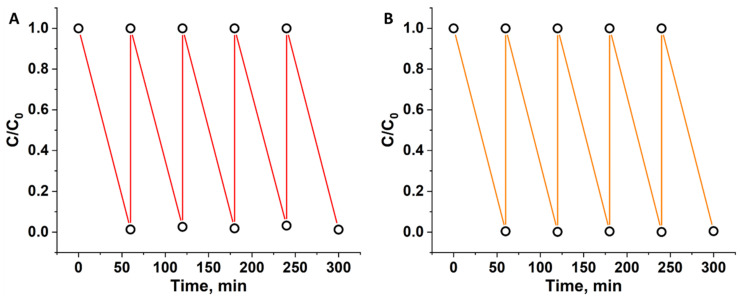
Cycling of 1^0.1^@TiO_2_ (**A**) and 2^0.1^@TiO_2_ (**B**) in RhB photocatalytic decomposition.

**Figure 7 nanomaterials-12-04282-f007:**
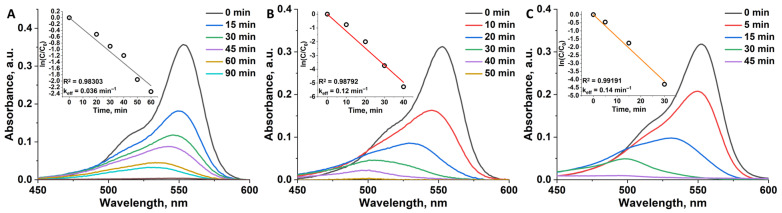
Absorption spectra of RhB solution before and after irradiation with sunlight (~30–35 mW cm^−2^) in the presence of TiO_2_ (**A**), 1^0.1^@TiO_2_ (**B**), and 2^0.1^@TiO_2_ (**C**) during different time intervals. Inserts are linear approximation of ln(C/C_0_) vs. time plots used for determination of k_eff_.

**Figure 8 nanomaterials-12-04282-f008:**
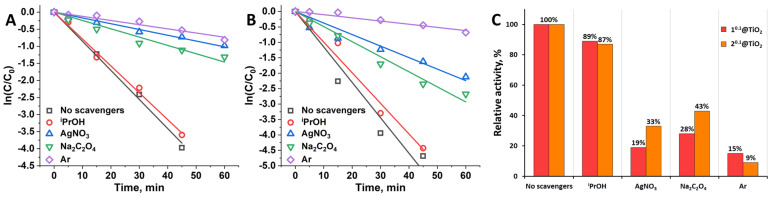
Effect of scavenger type on the rate constants of photocatalytic degradation of RhB in the presence of 1^0.1^@TiO_2_ (**A**) and 2^0.1^@TiO_2_ (**B**); Relative activity of the catalysts in the presence of different scavengers (**C**).

**Figure 9 nanomaterials-12-04282-f009:**
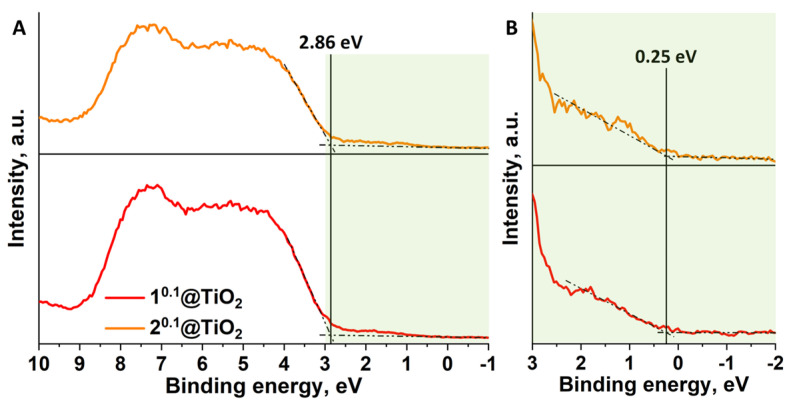
(**A**) Valence-band XPS spectra of n^0.1^@TiO_2_; (**B**) Enlarged part of the left spectra (colored in green).

**Figure 10 nanomaterials-12-04282-f010:**
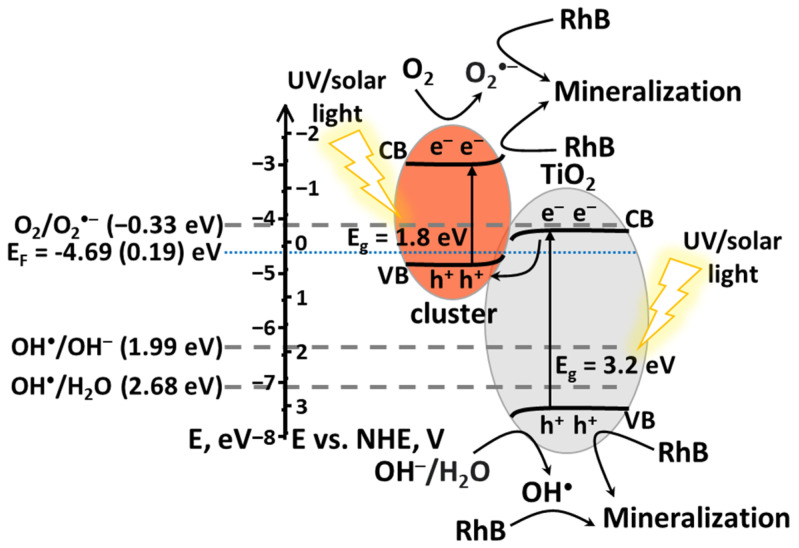
Schematic representation of S-type heterojunction photocatalytic mechanism for n^0.1^@TiO_2_.

**Table 1 nanomaterials-12-04282-t001:** The parameters of porous structure of samples under investigation.

Sample	Specific Surface Area, m^2^ g^−1^		V_pore_/cm^3^·g^−1^
	BET	DFT	
TiO_2_	253.2	277.0	0.406
1^0.1^@TiO_2_	213.9	178.5	0.382
2^0.1^@TiO_2_	213.2	186.6	0.349

**Table 2 nanomaterials-12-04282-t002:** Effective rate constants (k_eff_) of RhB photodegradation under UV light (λ = 365 nm, 13 mW cm^−2^).

Sample	Rate Constant (k_eff_), min^−1^	
TiO_2_	0.02	
	*n* = 1	*n* = 2
n^0.1^@TiO_2_	0.099	0.11
n^0.5^@TiO_2_	0.10	0.075
n^1^@TiO_2_	0.11	0.062
n^1.5^@TiO_2_	0.096	0.025
C_cat_ = 0.125 g L^−1^		
	*n* = 1	*n* = 2
n^0.1^@TiO_2_	0.055	0.026
C_cat_ = 0.375 g L^−1^		
	*n* = 1	*n* = 2
n^0.1^@TiO_2_	0.047	0.081

## Data Availability

The data that support the findings of this study are available from the corresponding author upon reasonable request.

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
