# Peer review of "Nano TiO2 and Molybdenum/Tungsten Iodide Octahedral Clusters: Synergism in UV/Visible-Light Driven Degradation of Organic Pollutants"

_nanomaterials, 2022, doi:10.3390/nano12234282_

Round 1

Reviewer 1 Report

This paper demonstrates an effective way of organic pollutants reduction by using TiO2, a typical electron transport material, which would be beneficial to the whole photo-catalysis process. The paper is sound and the novelty is appreciated. Therefore, I would like to recommend its publication after minor adjustments.

[1] TEM is a very interesting technology, so how about AFM images for TiO2 here? And some corresponding technical papers shall be cited: Adv. Energy Mater. 2022, 12, 2103977. and Energy Environ. Sci. 2022, 15, 2479.

[2] The XPS part can also be supported by correlated paper: Sci. China Chem. 2021, 64, 581.

[3] The photocatalysis can be enabled by traditional photovoltaics, so some high-PCE works shall be included as references: Nat. Mater. 2022, 21, 656., Adv. Mater. 2022, 34, 2202089. and ACS Energy Lett. 2022, 7, 2547.

[4]. The language of this paper can be further improved. [5]. Please also comment on the TiO2's effect versus SiO2.

Author Response

This paper demonstrates an effective way of organic pollutants reduction by using TiO2, a typical electron transport material, which would be beneficial to the whole photo-catalysis process. The paper is sound and the novelty is appreciated. Therefore, I would like to recommend its publication after minor adjustments.

Answer: Thank you for your positive feedback and relevant suggestions, which we address below.

  1. TEM is a very interesting technology, so how about AFM images for TiO2 here? And some corresponding technical papers shall be cited: Adv. Energy Mater. 2022, 12, 2103977. and Energy Environ. Sci. 2022, 15, 2479.

Answer. Thank you for your valuable comment. Indeed, the AFM technique can provide additional data on the particle size and size distribution in TiO2 and nx@TiO2. Nevertheless, we believe, that combination of TEM and XRPD (size determined using Scherrer equation) provides solid and reliable data on size and morphology of the particles, and utilization of AFM is excessive for this work.

  1. The XPS part can also be supported by correlated paper: Sci. China Chem. 2021, 64, 581.

Answer. Suggested references were added in section “3.3 X-ray photoelectron spectroscopy (XPS)”.

  1. The photocatalysis can be enabled by traditional photovoltaics, so some high-PCE works shall be included as references: Nat. Mater. 2022, 21, 656., Adv. Mater. 2022, 34, 2202089. and ACS Energy Lett. 2022, 7, 2547.

Answer. Suggested references were added in Introduction section.

  1. The language of this paper can be further improved.

Answer. The language was improved.

  1. Please also comment on the TiO2's effect versus SiO2.

Answer. Earlier, we studied the effect of pure SiO2 and cluster-containing SiO2 in RhB photodegradation (unpublished results). Pure SiO2 is known not to possess photocatalytic properties due to high Eg value. All cluster-containing samples also demonstrated very low efficiency (~5% of dye degradation after 90 min of irradiation).

Reviewer 2 Report

In the rewieved manuscript the titanium dioxide was obtained by direct aqueous hydrolysis of titanium (IV) isopropoxide and then impregnated with aqueous solutions of molybdenum/tungsten iodide octahedral clusters to increase its photocatalytic activity. The impregnation not affected the morphology of the titanium particles. Moreover, the stability of the obtained materials was preserved for up to 5 cycles of photodegradation.

Below, I have listed the weak points of the manuscript:

1. The iodide octahedral clusters were synthesized according to known procedure (ref. 24),

2. The titanium dioxide was obtained by the hydrolysis of titanium (IV) isopropoxide. Here, the relevant literature reference(s) is(are) needed.

In the light of the above facts, the novelty of this experimental work is rather low (in my opinion). Beside this the manuscript is quite well written. Also discussion of the obtained results is properly conducted.

To conclude, the originality of the reviewed manuscript is rather low but it can be published in Nanomaterials Journal.

Author Response

In the rewieved manuscript the titanium dioxide was obtained by direct aqueous hydrolysis of titanium (IV) isopropoxide and then impregnated with aqueous solutions of molybdenum/tungsten iodide octahedral clusters to increase its photocatalytic activity. The impregnation not affected the morphology of the titanium particles. Moreover, the stability of the obtained materials was preserved for up to 5 cycles of photodegradation.

Answer: Thank you for your positive feedback and relevant suggestions, which we address below.

Below, I have listed the weak points of the manuscript:

  1. The iodide octahedral clusters were synthesized according to known procedure (ref. 24),
  2. The titanium dioxide was obtained by the hydrolysis of titanium (IV) isopropoxide. Here, the relevant literature reference(s) is(are) needed.

Answer: Relevant references were added.

In the light of the above facts, the novelty of this experimental work is rather low (in my opinion). Beside this the manuscript is quite well written. Also discussion of the obtained results is properly conducted.

To conclude, the originality of the reviewed manuscript is rather low but it can be published in Nanomaterials Journal.

Reviewer 3 Report

This manuscript reported the titanium impregnated with dioxide (TiO2) octahedral cluster complexes [{M6I8}(DMSO)6](NO3)4 (M = Mo, W) for photocatalytic degradation of Rhodamine B (RhB). TiO2 particles were prepared by the hydrolysis of titanium (IV) isopropoxide, and the cluster-containing materials nx@TiO2 were obtained by impregnating TiO2 with [{Mo6I8}(DMSO)6](NO3)4 or [{W6I8}(DMSO)6](NO3)4 octahedral clusters, respectively. The morphology and elemental composition of nx@TiO2 were studied. The specific surface area, pore structure and absorption/emission spectra of nx@TiO2 were also measured. Based on the degradation rate of RhB, the photocatalytic activity of nx@TiO2 was investigated. The good stability of nx@TiO2 was confirmed by cyclic experiments. Furthermore, the photocatalytic activity of nx@TiO2 was tested under daylight conditions. The active species during the photodegradation of RhB were also studied, the formation of S-heterojunction was confirmed, and the propose mechanism of dye photodegradation was discussed.
I think this manuscript is suitable for publication in Nanomaterials, provided the authors consider the following comments.
(1)The authors mentioned that “Modern methods for the removal of such pollutants, such as coagulation, precipitation and adsorption, have several disadvantages forcing researchers to search for alternative methods” (Page 1). It is necessary to provide the disadvantages of the existing pollutant removal methods (such as coagulation, precipitation and adsorption) and highlight the advantages of nx@TiO2 photocatalyst developed in this manuscript.
(2)The dose of photocatalyst may affect the degradation behavior of pollutants. The authors needs to investigate the effect of nx@TiO2 dose on RhB degradation behavior by relevant experiments.
(3)In this manuscript, 10.1@TiO2 (A) and 20.1@TiO2 were selected as representatives for testing and data analysis, but the author did not explain why 10.1@TiO2 (A) and 20.1@TiO2 were selected as representatives. Please make corresponding supplement and modification.
(4)The authors mentioned that “the materials obtained here demonstrate high efficiency both under UV and sunlight irradiation and, therefore, they can be used for efficient wastewater treatment from organic pollutants”(Page 13). The authors need to illustrate the potential of nx@TiO2 photocatalysts for wastewater treatment by comparing the dye-degradation efficiency of other works.

Author Response

This manuscript reported the titanium impregnated with dioxide (TiO2) octahedral cluster complexes [{M6I8}(DMSO)6](NO3)4 (M = Mo, W) for photocatalytic degradation of Rhodamine B (RhB). TiO2 particles were prepared by the hydrolysis of titanium (IV) isopropoxide, and the cluster-containing materials nx@TiO2 were obtained by impregnating TiO2 with [{Mo6I8}(DMSO)6](NO3)4 or [{W6I8}(DMSO)6](NO3)4 octahedral clusters, respectively. The morphology and elemental composition of nx@TiO2 were studied. The specific surface area, pore structure and absorption/emission spectra of nx@TiO2 were also measured. Based on the degradation rate of RhB, the photocatalytic activity of nx@TiO2 was investigated. The good stability of nx@TiO2 was confirmed by cyclic experiments. Furthermore, the photocatalytic activity of nx@TiO2 was tested under daylight conditions. The active species during the photodegradation of RhB were also studied, the formation of S-heterojunction was confirmed, and the propose mechanism of dye photodegradation was discussed. I think this manuscript is suitable for publication in Nanomaterials, provided the authors consider the following comments.

Answer: Thank you for your positive feedback and relevant suggestions, which we address below.

  1. The authors mentioned that “Modern methods for the removal of such pollutants, such as coagulation, precipitation and adsorption, have several disadvantages forcing researchers to search for alternative methods” (Page 1). It is necessary to provide the disadvantages of the existing pollutant removal methods (such as coagulation, precipitation and adsorption) and highlight the advantages of nx@TiO2 photocatalyst developed in this manuscript.

Answer. The advantages of advanced oxidation process, which is related to all titania-based catalysts, are presented in Introduction section: “Such process has many advantages, such as complete mineralization of pollutants during the reaction, no need for regeneration after utilization, versatility, high efficiency even at low concentrations of pollutants, and compatibility with other methods”. Disadvantages of other modern methods were additionally provided.

  1. The dose of photocatalyst may affect the degradation behavior of pollutants. The authors need to investigate the effect of nx@TiO2 dose on RhB degradation behavior by relevant experiments.

Answer. Suggested experiments were conducted. Corresponding discussion was added to the main text.

  1. In this manuscript, 10.1@TiO2 (A) and 20.1@TiO2 were selected as representatives for testing and data analysis, but the author did not explain why 10.1@TiO2 (A) and 20.1@TiO2 were selected as representatives. Please make corresponding supplement and modification.

Answer. Explanation of 10.1@TiO2 (A) and 20.1@TiO2 was provided in the main text (see section “3.6 Photocatalytic degradation of RhB”, subsection “UV-light”) – “Since the highest catalytic efficiency along with the lowest cluster content was shown for n0.1@TiO2, these samples were chosen for further studies.”

  1. The authors mentioned that “the materials obtained here demonstrate high efficiency both under UV and sunlight irradiation and, therefore, they can be used for efficient wastewater treatment from organic pollutants” (Page 13). The authors need to illustrate the potential of nx@TiO2 photocatalysts for wastewater treatment by comparing the dye-degradation efficiency of other works.

Answer. The comparison of the efficiency of nx@TiO2 under UV-light to other TiO2-based photocatalysts was mentioned in section “3.6 Photocatalytic degradation of RhB”, subsection “UV-light”: “Taking into account experimental conditions, rate constant values of n0.1@TiO2 are comparable to the most efficient photocatalysts known in literature [36,37].”.
